# The Effect of Early Life Stress on Emotional Behaviors in GPR37KO Mice

**DOI:** 10.3390/ijms23010410

**Published:** 2021-12-30

**Authors:** Vandana Veenit, Xiaoqun Zhang, Antonio Ambrosini, Vasco Sousa, Per Svenningsson

**Affiliations:** Neuro Svenningsson, Department of Clinical Neuroscience, Karolinska Institutet, 171 76 Stockholm, Sweden; xiaoqun.zhang@ki.se (X.Z.); ambrosiniantonio@live.it (A.A.); vasco.sousa@ki.se (V.S.)

**Keywords:** limited nesting material, early life stress, P-T286-CaMKII, non-motor behaviors, corticosterone, hippocampus, amygdala, gender specific, context dependent

## Abstract

GPR37 is an orphan G-protein-coupled receptor, a substrate of parkin which is linked to Parkinson’s disease (PD) and affective disorders. In this study, we sought to address the effects of early life stress (ELS) by employing the paradigm of limited nesting material on emotional behaviors in adult GPR37 knockout (KO) mice. Our results showed that, while there was an adverse effect of ELS on various domains of emotional behaviors in wild type (WT) mice in a sex specific manner (anxiety in females, depression and context-dependent fear memory in males), GPR37KO mice subjected to ELS exhibited less deteriorated emotional behaviors. GPR37KO female mice under ELS conditions displayed reduced anxiety compared to WT mice. This was paralleled by lower plasma corticosterone in GPR37KO females and a lower increase in P-T286-CaMKII by ELS in the amygdala. GPR37KO male mice, under ELS conditions, showed better retention of hippocampal-dependent emotional processing in the passive avoidance behavioral task. GPR37KO male mice showed increased immobility in the forced swim task and increased P-T286-CaMKII in the ventral hippocampus under baseline conditions. Taken together, our data showed overall long-term effects of ELS—deleterious or beneficial depending on the genotype, sex of the mice and the emotional context.

## 1. Introduction

GPR37 is an orphan G-protein-coupled receptor which is highly expressed in the brain [1,2,3]. GPR37 is also called parkin-associated endothelin B-like receptor (PAEL-R), owing to its connection with parkin and its homology with the endothelin B receptor [4]. GPR37 can support or hinder neuronal viability depending upon its folding and cellular localization [5,6]. The surface expression of the receptor exhibits neuroprotective properties while the intracellular retention of the protein causes misfolding, aggregation, and degeneration [4]. Additionally, forced overexpression of the receptor leads to profound neurodegeneration in animal models, which shows selectivity for dopaminergic (DA) neurons [7]. Studies focusing on the dopaminergic system in GPR37 knockout (KO) mice demonstrate a decrease in dopamine content in the striatum, dysregulated dopaminergic signaling in the brain and specific deficits in motor behavior sensitive to nigrostriatal dysfunction [8,9,10,11].

GPR37 is also linked to affective disorders [12,13,14,15]. GPR37 is downregulated in the dorsolateral prefrontal cortex and anterior cingulate (brain regions associated with hedonism, impulse control, working memory and depression) in major depressive disorder (MDD) and upregulated in bipolar disorder [15]. The possible interaction between GPR37 expression and affective disorders is also related to Parkinson’s disease (PD), as PD patients often exhibit non-motor symptoms like mood- and memory-related disorders [16]. Additionally, anxiety and depression have remarkably prominent level of comorbidity in PD patients [17], much higher than in depressed patients without PD [18]. Studies investigating emotional symptoms in GPR37KO mice are few and have been done with different age groups, which makes it difficult to compare results across studies. For example, one study [12] reports a reduction in the startle response to acoustic stimuli in adult GPR37KO mice in both genders (4–6 months old), and increased anxiety- and depression-like behaviors in aged female GPR37KO mice (16–18 months old), whereas another study [19] reports less anxiety in young adult GPR37KO male mice (2 months old).

A growing body of evidence suggests that adverse experiences during childhood are associated with an increased risk of development of psychiatric disorders in adulthood [20]. To gain insight into the neurobiology of depression, animal models of early life stress (ELS) have been developed in primates and rodents [21,22]. However, not all individuals who experience stressful events in their life become depressed. Stressful events usually lead to depression in individuals with a genetic predisposition, thereby suggesting that depression is commonly precipitated by the interaction of environmental factors with an underlying genetic component. There is a growing body of evidence that genes implicated in neurodegeneration influence early brain development. In this paper, we aimed to address the effect of early life adversity on the non-motor behaviors in GPR37KO mice compared to wild-type (WT) mice. Given the association of GPR37 with Parkinson disease [23] and based on the evidence showing the link between abnormal Ca^2+^-calmodulin independent protein kinase II (CaMKII) function with motor deficits and synaptic deficits in experimental parkinsonism [24], we examined protein levels of phosphorylated alpha CaMKII-(P)-T286-CaMKIIand total CaMKII in the hippocampus and amygdala of these mice in the aftermath of all the behavioral tests. Similarly, based on the role of the hypothalamic-pituitary adrenal axis in mediating the effects of early life stress and its link with emotional behavior, we also examined the levels of terminal corticosterone in these mice [25,26,27,28].

Here, we report that the absence of GPR37 affects non motor behavior in a gender-specific manner as has been reported earlier [12]. GPR37KO mice showed resilience towards ELS in a gender- and context-dependent manner.

## 2. Results

### 2.1. The Schematic of the Experimental Design

Figure 1 shows the experimental design in which mice WT or GPR37KO mice underwent early life stress followed by a battery of behavioral tests in adulthood before being sacrificed for biochemical analyses.

### 2.2. Effect of ELS on Anxiety-like Behaviors in Adult GPR37KO Mice

In order to evaluate anxiety levels, we carried out an elevated plus maze test. A three-way ANOVA on the number of visits to open arms revealed a significant effect of ELS [F (1,57) = 4.780, *p* < 0.05] with no significant effects observed for sex or genotype (Appendix A). Post hoc analysis using the two-stage linear step-up procedure of Benjamini, Krieger and Yekutieli revealed that the WT ELS female group condition exhibited a reduced frequency of entering the open arms compared to the WT no-ELS female group (Figure 2A, *p* < 0.05), while KO ELS females showed a tendency to make more visits to open arms compared to WT ELS females (Figure 2A, *p* < 0.1 > 0.05). A three-way ANOVA on the percentage of time spent in open arms revealed significant effect of sex x ELS x genotype interaction [F (1,56) = 4.871, *p* < 0.05] (Appendix A). Post hoc analysis showed that GPR37KO ELS females spent more time in open arms compared to GPR37KO ELS males and WT ELS females; WT ELS females spent less time in open arms compared to WT no-ELS females (Figure 2B, *p* < 0.05). KO ELS females tend to spend more time in open arms compared to KO no-ELS females (Figure 2B, *p* < 0.1 > 0.05).

We also assessed the level of anxiety using a light-dark box test. A three-way ANOVA on the frequency of entering the light chamber revealed a significant effect of sex x genotype [F (1,59) = 12.29, *p* < 0.05] with no effect observed for ELS (Figure 3A) (Appendix A). Post hoc analysis revealed that WT no-ELS males exhibited a higher frequency of entering the light chamber than GPR37KO no-ELS males; WT females exhibited a reduced frequency of entering the light chamber than GPR37KO females under any treatment condition; WT no-ELS males and WT ELS males made more visits to the light chamber than WT no-ELS females and WT ELS females respectively, while KO no-ELS males made fewer visits to the light chamber than KO no-ELS females (Figure 3A, *p* < 0.05). A three-way ANOVA for percentage of time spent in the light chamber showed significant effects of sex [F (1,57) = 8.16, *p* < 0.05], and genotype x ELS [F (1,57) = 8.67, *p* < 0.05], with no significant effect observed for other factors (Appendix A). Post hoc analysis showed that WT ELS females spent less time in the light chamber compared to WT ELS males, WT no-ELS females and GPR37KO ELS females (Figure 3B, *p* < 0.05). Taken together, there was an effect of ELS in females with increased anxiety levels being observed in WT ELS female compared to WT no-ELS female. GPR37KO females displayed less anxiety compared to GPR37KO males and WT females especially under ELS treatment condition suggesting that deletion of GPR37 might confer anxiolytic action in females.

### 2.3. Effect of ELS on Emotional Memory Behavior in Adult GPR37KO Mice

A passive avoidance task was carried out to evaluate emotionally loaded memory processing. A three-way ANOVA revealed sex differences in the training phase of the passive avoidance task, with females exhibiting a short latency to enter the dark compartment [F (1,61) = 12.33, *p* < 0.05, Figure 4A] (Appendix A). In the retention phase of the passive avoidance task, a three-way ANOVA revealed the significance of genotype [F (1,60) = 6.89, *p* < 0.05] and ELS [F (1,60) = 4.612, *p* < 0.05] (Appendix A). Post hoc analysis revealed that GPR37KO ELS males exhibited a higher latency to enter the dark compartment compared to WT ELS males (Figure 4B, *p* < 0.05); GPR37KO no-ELS males displayed a tendency towards higher latency to enter the dark compartment compared to WT no-ELS (*p* < 0.1 > 0.05). WT ELS males displayed a tendency towards lower latency to enter the dark compartment compared to WT no-ELS males (Figure 4B, *p* (*p* < 0.1 > 0.05).

The results of the passive avoidance task showed that GPR37KO males exhibited a tendency to display better retention for aversive memory compared to WT males. WT ELS males showed a tendency of poor retentive memory compared to WT no-ELS males.

### 2.4. Effect of ELS on Depression-like Behavior in Adult GPR37KO Mice

A forced swim test was conducted to evaluate “active coping” vs. “passive coping” strategies in an adverse situation [29,30] and aspects of depression-like behavior [31]. A three-way ANOVA revealed a significant effect of genotype [F (1,57) = 5.63, *p* < 0.05], and a trend towards significance for sex x ELS interaction [F (1,57) =3.108, *p* < 0.1 > 0.05] (Appendix A). Post hoc analysis showed that GPR37KO no-ELS males exhibited a greater time spent immobile compared to the WT no-ELS males; WT no-ELS females exhibited a greater time spent immobile than the WT no-ELS males; WT ELS males exhibited a tendency to spend more time immobile compared to WT no-ELS males (Figure 5, *p* = 0.08).

The data of the Forced swim test suggests that GPR37KO no-ELS males exhibited higher depression like behavior compared to their WT no-ELS counterparts. WT ELS males tended to be more depressed than WT no-ELS males.

### 2.5. Effect of ELS on Body Weight in GPR37KO Mice

When mice were examined for weight on P12, a two-way ANOVA revealed a significant effect of genotype [F (1,58) = 5.064, *p* < 0.05; treatment F (1, 58) = 15.54, *p* < 0.05] with no effect observed for genotype x treatment interaction (Figure 6A) (Appendix A). Post hoc analysis using the two-stage linear step-up procedure of Benjamini, Krieger and Yekutieli revealed that WT ELS and GPR37KO ELS groups exhibited reduced body weight compared to WT no-ELS and GPR37KO no-ELS respectively (Figure 6A). In adulthood, a three-way ANOVA revealed a significant effect of sex [F (1,67) = 142.7, *p* < 0.05, treatment F (1,67) = 8.612, *p* < 0.05 and sex x genotype interaction F (1,67) = 6.489, *p* < 0.05] (Appendix A). Post hoc analysis using the two-stage linear step-up procedure of Benjamini, Krieger and Yekutieli revealed that WT ELS females exhibited reduced body weight compared to WT no-ELS females (Figure 6B, *p* < 0.05); GPR37KO males subjected to ELS tended to weigh less compared to GPR37KO no-ELS males (Figure 6B, *p* < 0.1 > 0.05).

Taken together, GPR37KO mice such as WT mice displayed reduced body weights in the immediate aftermath of ELS and the treatment effect persisted until adulthood in WT females and GPR37KO males.

### 2.6. Effect of ELS on Corticosterone Level in Adult GPR37KO Mice

The level of the stress hormone corticosterone was examined in the mice subjected to behavior tests and mice under basal conditions. A three-way ANOVA on corticosterone measures revealed significant effects of ELS [F (2,65) = 137.2, *p* < 0.0001], sex [F (1,65) = 259.8, *p* < 0.0001], genotype [F (1,65) = 5.456, *p* < 0.05], ELS x sex [F (2,65) = 10.96, *p* < 0.0001], sex x genotype [F (2,65) = 137.2, *p* < 0.05], and ELS x sex x genotype [F (2,65) = 137.2, *p* < 0.01]. Post hoc analysis revealed that females of all the groups except GPR37KO basal females exhibited higher corticosterone level than their corresponding male groups (Figure 6, *p* < 0.05). Mice subjected to behavioral tests had higher corticosterone levels than the mice in the basal group within the same genotype in both sexes (Figure 7, *p* < 0.05). GPR37KO basal females exhibited lower corticosterone than the WT basal females (Figure 7, *p* < 0.05); GPR37KO no-ELS males exhibited reduced corticosterone level compared to WT no-ELS males (Figure 7, *p* < 0.05).

Corticosterone data showed that GPR37KO basal females displayed reduced corticosterone level compared to WT basal females, GPR37KO ELS females displayed reduced corticosterone level compared to WT ELS females, and mice subjected to a battery of behavior tests displayed higher levels of corticosterone. Females had higher levels of corticosterone than males.

### 2.7. Effect of ELS on P-T286-CaMKII in DH, VH and Amygdala, in GPR37KO Mice

CaMKII is an important member of the calcium/calmodulin-activated protein kinase family, functioning in neural synaptic stimulation. We examined the level of P-T286-CaMKII in important brain regions associated with learning, memory and emotional processing. A three-way ANOVA did not reveal any clear-cut significant effects of genotype, ELS, sex, or their interactions in the DH (Appendix A).

However, there was a trend towards significance for some factors in the ANOVA test, such as sex [F (1,64) = 3.619, *p* = 0.061] and treatment x genotype [F (2,64) = 2.542, *p* = 0.086. Post hoc analysis revealed that WT no-ELS males exhibited a reduced density of P-T286-CaMKII/total CaMKII compared to WT no-ELS females; WT ELS males exhibited a tendency towards reduced density of P-T286-CaMKII/total CaMKII compared to WT ELS females (*p* < 0.1 > 0.05); GPR37KO males under basal condition exhibited a higher density of P-T286-CaMKII/total CaMKII compared to GPR37KO ELS males (Figure 8A, *p* < 0.05). In the VH, a three-way ANOVA revealed a significant effect of ELS x sex on the density level of P-T286-CaMKII/total CaMKII [F (2,61) = 3.193, *p* < 0.05] and a trend towards significance for ELS x sex x genotype F (2,61) = 3.109, *p* = 0.051 (Appendix A). Post hoc analysis revealed that GPR37KO males under basal conditions exhibited a higher density of P-T286-CaMKII/total CaMKII compared to WT males under basal conditions; GPR37KO males under basal conditions exhibited higher density of P-T286-CaMKII/total CaMKII compared to GPR37KO females under basal conditions; GPR37KO no-ELS females and GPR37KO ELS females exhibited higher densities of P-T286-CaMKII/total CaMKII compared to GPR37KO females under basal conditions (Figure 8B, *p* < 0.05). A three-way ANOVA revealed a significant effect of ELS in the amygdala [F (2,64) = 5.977, *p* < 0.05] (Appendix A). Post hoc analysis showed that WT no-ELS females exhibited reduced density of P-T286-CaMKII/total CaMKII compared to the GPR37KO no-ELS females; GPR37KO basal females exhibited reduced density of P-T286-CaMKII/total CaMKII compared to the GPR37KO no-ELS females (Figure 8C, *p* < 0.05).

Taken together, GPR37KO no-ELS males exhibited increased densities of P-T286-CaMKII in the VH compared to WT no-ELS males; behavior tests increased P-T286-CaMKII density in the amygdala in GPR37KO no-ELS females compared to WT no-ELS females.

### 2.8. Effect of ELS on Total CaMKII in DH, VH and Amygdala, in GPR37KO Mice

We examined the level of total CaMKII in important brain regions associated with learning, memory and emotional processing. A three-way ANOVA on total CaMKII did not show any effects of sex, genotype, ELS or interactions between these factors. The mean value of total CaMKII in various regions are summarized below in Table 1:

## 3. Discussion

In this study, we set out to examine the effect of early life stress in PD-related GPR37KO mice compared to WT mice, using both male and female mice. Although there are studies which have examined non-motor behaviors in the GPR37KO mice, it is hard to compare the results across studies as they have been conducted under different study designs and with different age groups of mice [12,19]. Our results suggest that GPR37KO mice display altered behavior compared to WT mice depending on the context and sex. The differences observed in behavior here cannot be attributed to differences in locomotion activity between sexes or genotype, as the total distance travelled in the EPM test (data not shown in the manuscript) was not different between male and female mice or between GPR37KO and WT mice (in 3–4-month-old adult mice used in this study).

Lower levels of GPR37 have been reported in depressed patients [15]. Similarly, under baseline conditions, GPR37KO males displayed a tendency to spend more of their time immobile, indicating increased passive coping strategies compared to WT mice, with no similar effect observed in females. While there has been a debate in the field regarding the validity of FST as a test for depression [29,30], time spent immobile in the FST can be interpreted more as a “passive coping strategy” than “learned helplessness” considering the latest discussions in the field [29,30]). Male GPR37KO mice displayed a tendency towards better retention of aversive memory as observed in passive avoidance tasks, with no such effect observed in female mice, suggesting they are better at retaining the memory of emotionally laden experiences.

Based on the readouts in anxiety tests, such as percentage of time spent in open arms during the EPM, number of visits to the light-dark box and the percentage time spent in the light chamber of the light-dark box, ELS exerts a protective effect in GPR37KO females. Although, not consistent across all the parameters of anxiety tests employed in this study, some parameters such as the number of visits to the light chamber highlight sex-related differences in anxiety levels. There are studies which report sex differences on the parameters of anxiety and depression [32,33,34,35].

Although many studies have examined emotional and cognitive behaviors in mouse models linked with PD [36,37], this is the first study to look at the effect of ELS on the emotional and cognitive domains in a genetic mouse model implicated in PD. Our data suggests that the effect of ELS on measures of anxiety and depression was not so consistent and strong in WT mice in this study. Similar cases have been encountered in another study where the effect of ELS is more pronounced in light-dark box tests but very mild in the EPM [28]. Our findings also show that the long-term effects of ELS can be deleterious or beneficial depending on its interaction with the genotype and sex of the mice. For example, the results of the anxiety tests (EPM, light-dark box) suggest that the absence of GPR37 confers protection in females under ELS conditions. Similarly, GPR37KO ELS males have better memory retention compared to WT no-ELS males. Chronic stress and elevated levels of glucocorticoids are risk factors for depression, and depression is a common non-motor symptom in PD patients. A recent study has shown that corticosterone administration accelerates the pathology of PD in a model of PD [26]. In the current study, female mice exhibited higher corticosterone levels than male mice under all treatment conditions, suggesting heightened stress responses in females compared to males. Similar findings have been reported in the literature [25,27]. Moreover, there was a significant decrease in the level of corticosterone in GPR37KO females under basal and ELS conditions compared to WT females. This might suggest attenuated hypothalamic pituitary adrenal axis in GPR37KO females and might explain the protective effects of the deletion of GPR37 observed in females under ELS conditions against anxiety, given the positive association of corticosterone with anxiety [38]. There is a clear-cut effect of ELS on body weight in the aftermath of early life stress, as shown by the body weight data on P12. However, this effect of ELS on body weight persisted only in WT ELS females in adulthood and not in other groups. GPR37KO pups displayed reduced body weight compared to WT mice on PND 12, suggesting a more severe physiological effect of ELS in GPR37KO mice, while the effect of ELS persisted on body weight in WT females in adulthood.

We decided to look at the expression of P-T286-CaMKII (a key synaptic signaling molecule that facilitates learning and memory processes) in important brain regions, the DH, VH and amygdala, which are involved in emotional and cognitive processing [39,40]. We chose to look at P-T286-CaMKII in this study over other potential plasticity markers as there are studies suggesting a link of P-T286-CaMKII with parkin [24]. We examined both the DH and VH separately because of their different roles in memory and emotional processing [41,42,43,44]. Our result showed that GPR37KO males have a higher density of P-T286-CaMKII in the VH compared to WT males under basal conditions. This finding might better explain the memory for aversive stimuli in GPR37KO males compared to wild type males as lesions in the VH have been reported to reduce fear expression [44]. Higher P-T286-CaMKII density in the hippocampus is associated with higher long-term potentiation LTP [45], and hence, better memory. GPR37KO females exposed to a battery of behavior tests exhibited lower density of P-T286-CaMKII in the amygdala compared to GPR37KO females under basal conditions and GPR37KO females exposed to ELS. GPR37KO males exposed to a battery of behavior tests have elevated densities of P-T286-CaMKII in the amygdala compared to WT mice, suggesting stress induced the higher density of P-T286-CaMKII in the amygdala. While corticosterone results and P-T286-CaMKII data may partly explain some of the behavior results observed in this study, it is likely that other molecular mediators are involved in producing the observed behavioral phenotypes and physiological parameters in these mice. Some of the studies have focused on the level of neurotransmitters in GPR37KO mice [12] or adenosinergic system [19] for explaining some of the behavioral phenotype.

## 4. Materials and Methods

### 4.1. Subjects and Housing

The experiments were approved by the local ethical committee at Karolinska Institute (N139-16) and conducted in accordance with the European Communities Council Directive of 24 November 1986 (86/609/EEC). Mice were housed in temperature- and humidity-controlled rooms (20 °C, 53% humidity) in well ventilated racks with a 12 h dark/light cycle. They had access to standard lab pellets and water ad libitum.

### 4.2. Breeding and Genotype

GPR37KO embryos on a C57BL/6J genetic background [46,47] were a kind gift from Dr. Randy Hall. Mice were subsequently bred and maintained in the Wallenberg animal facility of Karolinska Institutet. The first breeding was between GPR37KO male x C57BL6J and then heterozygote breeding was carried out until the experiments. Mice were genotyped to confirm genetic deletion of GPR37.

Genotyping was performed as previously described [11]. DNA for genotyping was extracted from the tail or by ear biopsy. The primers used are listed in Table 2. Briefly, DNA (2 µL) was amplified on a T100™ Thermal Cycler (#1861096, Bio-Rad Laboratories, Inc., Hercules, CA, USA) for 35 cycles (95 °C for 60 s, 58 °C for 30 s, and 72 °C for 60 s). After additional incubation at 72 °C for 10 min and being cooled to 4 °C, PCR products were subjected to electrophoresis in 1.5% agarose gel in GelRed. The relative intensity of the PCR bands was analyzed using the ChemiDoc™ MP Imaging System (#12003154, Bio-Rad Laboratories, Inc.)

### 4.3. General Experimental Plan

Eight to twelve-week-old male and female mice (WT and GPR37KO) were used. WT × WT and GPR37KO × GPR37KO mice were bred to generate experimental mice. Female mice were housed singly upon confirmation of pregnancy. Once the female gave birth, four experimental groups were assigned — WT no-ELS, WT + ELS, GPR37KO no-ELS and GPR37KO + ELS. On postnatal day (PND) 2, cages were changed for all the groups. For the WT no-ELS and GPR37KO no-ELS groups, a little bit of bedding, tube, and nesting material were transferred to the new cage along with the new nesting material. Nesting material consisted of one square piece of cotton material measuring 5 cm × 5 cm. This material is shredded by the dam to create a nest area. For the ELS groups, 1/4th of the nesting material compared to the control group was used. On PND2, we measured the weight of the mother and made a note of the number of pups (male and female). The number of pups per mother was 6 to 8 (males and females). The cages were left undisturbed from PND2 to PND9. However, we examined the behavior of the mothers (to check that they did not eat pups) without disturbing the cage. On PND9, the ELS protocol finished, and the mice (pups and the mother) were transferred to a standard cage. Cages were also changed for the other two groups (WT, GPR37KO), who were not subjected to the paradigm of ELS. Pups were weaned on PND21. Male and female littermates were placed in different cages. The number of littermates in a cage was kept 4–5 for males and 5–6 for females. The mice for all the groups were selected from 3–4 litters. At the time of weaning, 5–8 males and females from non ELS groups (from different litters) were assigned as basal groups (both WT and GPR37KO). Eventually, there were six groups — WT basal, WT no-ELS, WT + ELS, GPR37KO basal, GPR37KO no-ELS and GPR37KO + ELS. A series of behavioral studies were carried out in adulthood in all the groups except the basal ones. Male and female mice were subjected to behavioral tests on different days in adulthood to rule out the possibility of pheromone effects on behavior. We did not assess the behavior in females at a particular estrous cycle, but we did ensure that the females were counterbalanced with respect to estrous cycles (assessed by vaginal smears prior to starting the series of behavior experiments) in all the groups to avoid any bias in the interpretation of results. Behavioral tests were carried out from the least to the most stressful to minimize previous test influence. Adequate resting days were given to the mice between tests. Behavioral experiments were performed between 9:00 and 16:00 h during the light phase of the circadian cycle. The number of subjects used in each experiment is specified in the appropriate Figure legend. The mice were euthanized 30 min after the last behavioral experiment, i.e., Forced Swim Test (FST), and their brains were later used for the Western Blot. Mice under basal conditions (WT basal and GPR37KO basal) were also sacrificed on the same day around the same time along with mice who underwent behavioral tests. The mice were sacrificed by decapitation, trunk blood was collected for corticosterone analysis and the brains were rapidly fresh-frozen in isopentane cooled in dry ice and then stored at −80 °C. The brains were subsequently used for western blot analysis by dissecting the dorsal and ventral hippocampus and the amygdala (described below). The schematic of the experimental design is shown in Figure 1.

### 4.4. Limited Nesting Material

The model of limited nesting material has been successfully used by a number of laboratories across the world to understand the mechanism of ELS-induced behavioral and neuroendocrine abnormalities [21]. The protocol of limited nesting material is described in the general experimental plan.

For mice undergoing the ELS protocol, the nesting material was placed on a fine-gauge aluminum mesh platform (mesh dimensions 0.4 cm × 0.9 cm, catalog no 57398; McNichols Co., Tempa, FL, USA), layered 2.5 cm above the cage floor. The cage floor was covered with a small amount of bedding (1/10th of the standard bedding) below the grid to absorb the urine [21]. This set-up permits the mouse droppings to fall below the platform without trapping the pups.

### 4.5. Behavioral Tests

#### 4.5.1. Elevated Plus Maze

The elevated plus maze was carried out to measure anxiety-like behavior [48]. The elevated plus maze apparatus consisted of an elevated platform (50 cm above the ground) with two opposing open arms (25 × 5 cm), two opposing closed arms (25 × 5 × 15 cm) and a central platform (5 × 5 cm). The light intensity was set to 14–15 lux in the open arms, 5–10 lux in the center and 3–4 lux in the closed arms. Mice were placed individually in the maze facing one of the closed arms. Mice could explore the apparatus for 5 min. Time spent in each arm was measured and analyzed by automated video tracking system (NOLDUS Ethovision XT11.5, Wageningen, The Netherlands).

#### 4.5.2. Light Dark Box

The light/dark exploration test was also conducted to assess anxiety-like behavior [49]. The light-dark box consisted of a dark compartment; 39 × 13 × 16 cm with a 13 × 8 cm aperture at floor level that opened onto a large white Plexiglas square arena (light compartment; 39 × 39 × 35 cm). In this test, mice were placed in a dark compartment, of the light-dark box. The light intensity in the light chamber was kept around 120 lux. The number of entries into the light compartment (defined as all 4 paws out of the shelter) and time spent inside the light compartment over a 15 min session was calculated by automated video-tracking system (NOLDUS Ethovision XT11.5). Mice that did not enter the light compartment were assigned a 900 sec latency to enter.

#### 4.5.3. Passive Avoidance Task

The passive avoidance test is used to assess fear related memory [50]. The passive avoidance apparatus (25 cm × 50 cm × 25 cm) consisted of two equal sized compartments connected by a sliding door (7 cm × 7 cm) (Ugo Basile, Comerio-Varese, Italy). The light intensity in the dark compartment was 2 lux and 250 lux in the bright compartment. The passive avoidance test comprised of training and testing. During training, the animal was placed in the bright compartment and could explore it for 60 s, after which the sliding door was opened, and the animal had a maximum of 300 s to enter the dark compartment. Once the mouse had entered the dark compartment, the sliding door was automatically closed and, after 3 s, a weak electrical stimulus (0.3 mA, 2 s scrambled current) was delivered through the grid floor. After 24 h, the animal was again gently placed in the light compartment, and the latency to enter the dark compartment was measured (retention latency) with a 540 s cutoff time. No electrical stimulus was given during the retention test. A significantly prolonged step-through latency during the test session indicates intact emotional contextual memory.

#### 4.5.4. Forced Swim Test

The forced swim test procedure was performed to evaluate depression-like behavior [31]. Each test session lasted for 6–7 min, during which the mice were individually placed in a vertical Plexiglas cylinder (height: 30 cm, diameter: 20 cm) filled to a 15 cm depth with water at 24–25 °C. The water was changed between every animal. The mice were removed from the water after 7 min and dried before they returned to the holding cage. Behavior was analyzed from minutes 2–6 of the test [51]. The experiment was recorded sideways and analyzed automatically using NOLDUS Ethovision XT11.5.

### 4.6. Brain Dissection

Frozen brains were sectioned using a cryostat and 150 µm-thick slices were mounted on slides and the regions of interest (dorsal hippocampal (DH), ventral hippocampus (VH) and amygdala) tissues dissected using a scalpel blade. A mouse atlas was used to ensure accurate dissection of the particular brain region. The tissue was collected in RNAse-free tubes.

### 4.7. Western Blot

Western blot using dorsal hippocampal, ventral hippocampal and amygdala samples was carried out according to standard protocol [52]. The samples were sonicated in 1% sodium dodecyl sulfate (SDS) containing phosphatase inhibitor and protease inhibitor and boiled for 5 min. The protein concentration in each sample was thereafter determined with a BCA-based kit (Thermo Fisher, Waltham, MA, USA). Twenty-five micrograms of each sample were re-suspended in a sample buffer and separated by SDS-PAGE using a 12% running gel and transferred to an Immobilon-P transfer membrane (Millipore). The membranes were incubated for 1 h at room temperature with 5% (*w*/*v*) bovine serum albumin in TBS-Tween 20. Primary antibodies from Cell signaling [(rabbit phospho CaMKII (Thr286), 1:1000) from rabbit phospho CaMKII (Thr286), 1:1000, Cell Signaling Technology 12716S, (mouse CaMKII-α (6G9), 1:1000) from Cell Signaling Technology 50049S, (mouse actin, 1:5000)] from Sigma-Aldrich A5441-100UL were used for phospho Thr286, CaMKII-α and actin, respectively. Secondary antibodies (goat anti-rabbit IRDye 800CW, 1:20,000) from Li-Cor 926-32211 were used for phospho Thr286, (goat anti-mouse IRDye 680RD, 1:20,000) from Li-Cor 926-68070 was used for CaMKII-α, and actin diluted in TBS-T containing 5% of BSA and 0.025% of sodium azide were applied overnight. All antibodies mentioned above are widely used, have been reported to be highly specific and do not cross-react. After three washing steps with TBS-T for 10 min each time, the membranes were submerged in fluorescently labeled secondary antibodies diluted in TBS-T containing 5% BSA and 0.01% SDS and incubated for 1 h at room temperature. After four washing steps in TBS-T and two washings in TBS, membranes were scanned in the appropriate channels (700 or 800 nm) using Odyssey CLx InfraRed Imaging system from LI-COR Biosciences (Lincoln, NE, USA). Quantification of the signals was done using software Image Studio 3.1. The background subtraction was carried out from the obtained intensity for each band, and all the values were normalized to actin, referred as a loading control. Immunoblotting was carried out with phosphorylation-state-specific antibodies against total CaMKII, P-T286-CaMKII and actin.

### 4.8. Corticosterone Assay

For measuring serum corticosterone level, trunk blood was collected from the mice after decapitation in a regular Eppendorf tube. After 30–40 min of coagulation, the samples were centrifuged at 11,000 rpm for 10 min. Serum (supernatant) was transferred to another Eppendorf tube and stored at −80 °C until analysis. Corticosterone was measured using a commercially available ELISA kit (ADI-901-097, Enzo Life Sciences, Farmingdale, NY, USA), following the manufacturer’s instructions. Serum samples were diluted 1:10 in the assay buffer.

### 4.9. Statistical Analysis

Data were analyzed with two-way or three-way ANOVAs as appropriate using the statistical package GraphPad Prism 9 (GraphPad software Inc., San Diego, CA, USA). Data were examined for normality using the Shapiro-Wilk test. If the data failed in the normality test, they were log-transformed and checked again for the normality test. Multiple comparisons were performed using the two-stage linear step-up procedure of Benjamini, Krieger and Yekutieli. All bars and error bars represent the mean ± SEM. Significance was set at *p* < 0.05, while the *p*-values were considered tending toward significance when 0.05 ≤ *p* ≤ 0.1. Graphs were created using GraphPad Prism 9.

## 5. Conclusions and Limitations

This study has addressed the effect of environmental manipulation on emotionally loaded behaviors in GPR37KO mice and reinforces that the absence of the GPR37 gene can be beneficial against the effects of stress in female mice. While the findings reported in this study show complex interaction between genes, sex and environment, the mechanism underlying this phenomenon warrants further investigation.

## Figures and Tables

**Figure 1 ijms-23-00410-f001:**
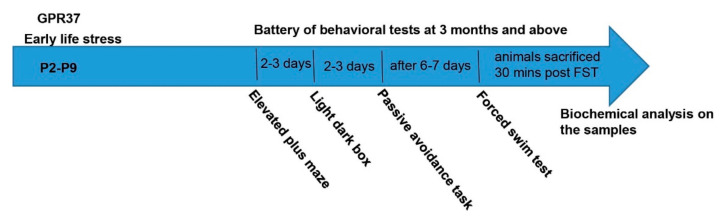
Schematic of the study showing the timelines of early life stress protocol and of the behavioral tests conducted in adulthood.

**Figure 2 ijms-23-00410-f002:**
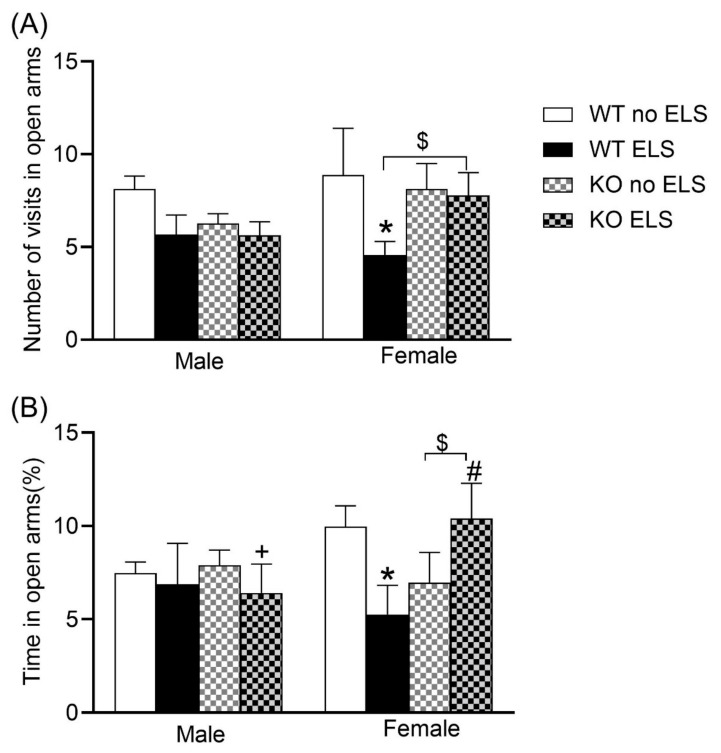
Effect of early life stress on WT and GPR37KO mice in elevated plus maze test. Frequency in open arms (**A**). Percent time spent in open arms (**B**). N/group: WT no-ELS male = 8, WT ELS male = 7, KO no-ELS male = 10, KO ELS male = 8, WT no-ELS female = 8, WT ELS female = 7, KO no-ELS female = 8 and KO ELS female = 9. All data represent mean ± SEM. * *p* < 0.05 treatment difference within genotype; ^#^
*p* < 0.05 versus corresponding WT; ^+^
*p* < 0.05 with respect to sex; ^$^
*p* > 0.05 < 0.1 calculated by three-way ANOVA followed by multiple comparisons using the two-stage linear step-up procedure of Benjamini, Krieger and Yekutieli. WT = Wild Type, KO = Knockout, ELS = Early Life Stress.

**Figure 3 ijms-23-00410-f003:**
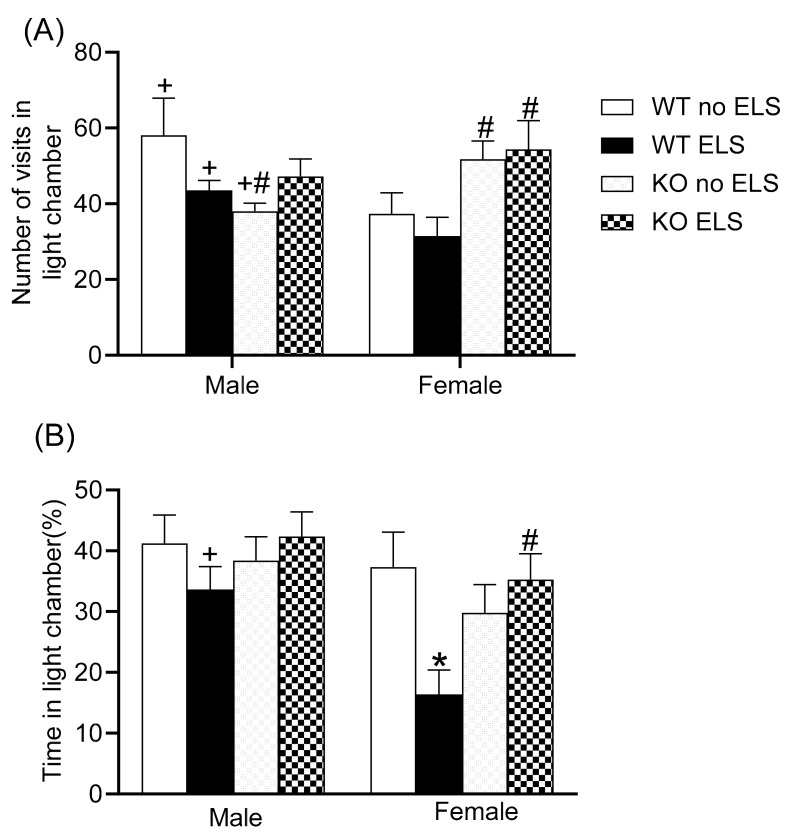
Effect of early life stress on WT and GPR37KO mice on light-dark box test. Frequency in light chamber (**A**). Percentage time spent in light chamber (**B**). N/group: WT no-ELS male = 8, WT ELS male = 7, KO no-ELS male = 10, KO ELS male = 10, WT no-ELS female = 8, WT ELS female = 7, KO no-ELS female = 9 and KO ELS female = 9. All data represent mean ± SEM. * *p* < 0.05 treatment difference within genotype; ^#^
*p* < 0.05 versus corresponding WT; ^+^
*p* < 0.05 with respect to sex; calculated by three-way ANOVA followed by multiple comparisons using the two-stage linear step-up procedure of Benjamini, Krieger and Yekutieli. WT = Wild Type, KO = Knockout, ELS = Early Life Stress.

**Figure 4 ijms-23-00410-f004:**
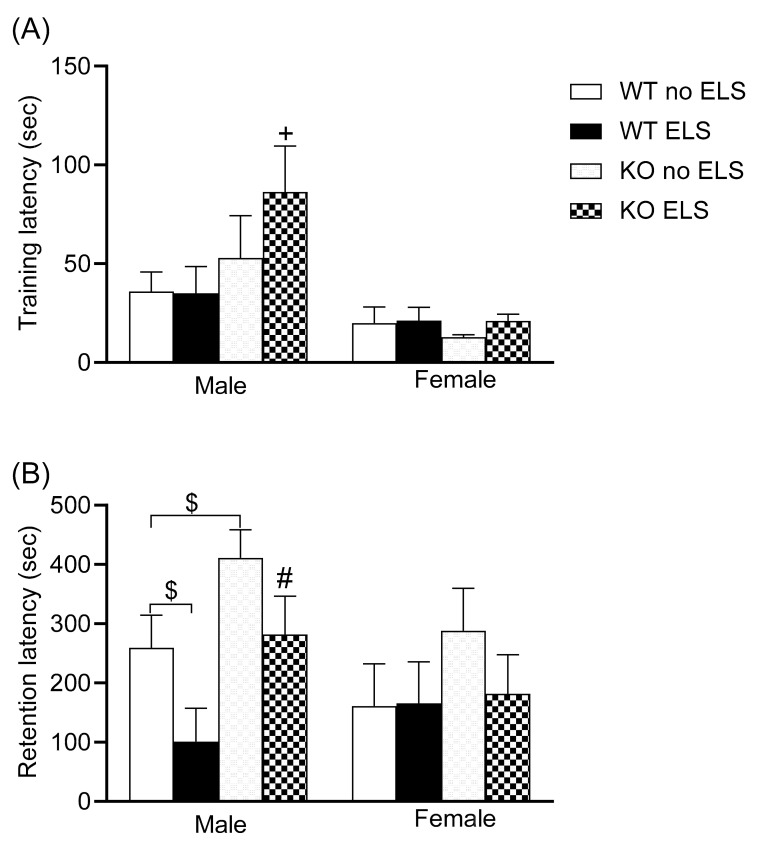
Effect of ELS on WT and GPR37KO mice in passive avoidance task. Training latency (**A**). Retention latency (**B**). N/group: WT no-ELS male = 9, WT ELS male = 7, KO no-ELS male = 9, KO ELS male = 10, WT no-ELS female = 8, WT ELS female = 7, KO no-ELS female = 9 and KO ELS female = 9. Results are expressed as mean ± SEM. All data represent mean ± SEM. ^#^
*p* < 0.05 versus corresponding WT; ^+^
*p* < 0.05 with respect to sex; ^$^
*p* > 0.05 < 0.1 calculated by three-way ANOVA followed by multiple comparisons using the two-stage linear step-up procedure of Benjamini, Krieger and Yekutieli. WT = Wild Type, KO = Knockout, ELS = Early Life Stress.

**Figure 5 ijms-23-00410-f005:**
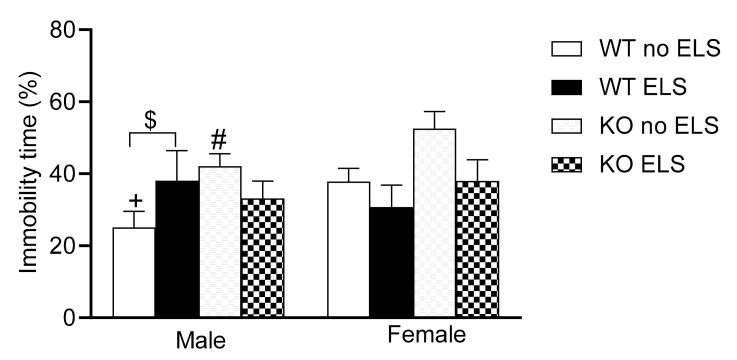
Effect of ELS on WT and GPR37KO mice on time spent immobile in forced swim task. N/group: WT no-ELS male = 8, WT ELS male = 7, KO no-ELS male = 9, KO ELS male = 9, WT no-ELS female = 8, WT ELS female = 7, KO no-ELS female = 8 and KO ELS female = 8. All data represent mean ± SEM. ^#^
*p* < 0.05 versus corresponding WT; ^+^
*p* < 0.05, with respect to sex; ^$^
*p* > 0.05 < 0.1 calculated by three-way ANOVA followed by multiple comparisons using the two-stage linear step-up procedure of Benjamini, Krieger and Yekutieli. WT = Wild Type, KO = Knock Out, ELS = Early Life Stress.

**Figure 6 ijms-23-00410-f006:**
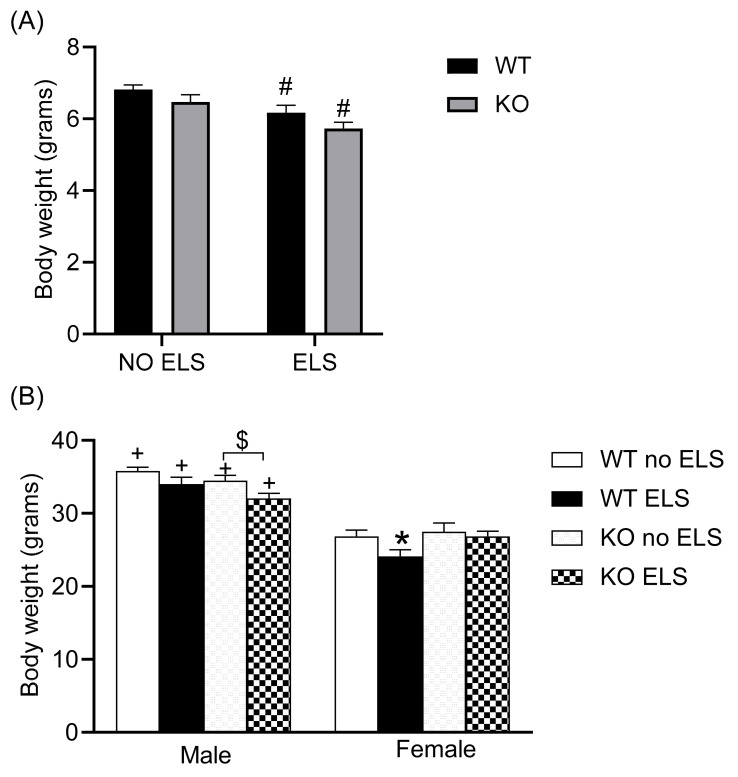
Effect of early life stress in WT and GPR37KO mice on body weight (BW). BW at P12 (**A**), N/group: WT no-ELS = 22, WT ELS = 16, KO no-ELS = 14, KO ELS = 11. BW of adult mice (**B**), N/group: WT no-ELS male = 10, WT ELS male = 10, KO no-ELS male = 10, KO ELS male = 7, WT no-ELS female = 11, WT ELS female = 7, KO no-ELS female = 12 and KO ELS female = 8. All data represent mean ± SEM. * *p* < 0.05 treatment difference within genotype, ^#^
*p* < 0.05 versus corresponding WT; ^+^
*p* < 0.05, with respect to sex, ^$^
*p* > 0.05 < 0.1 calculated by two-way ANOVA/three-way ANOVA followed by multiple comparisons using two-stage linear step-up procedure of Benjamini, Krieger and Yekutieli. WT = Wild Type, KO = Knockout, ELS = Early Life Stress.

**Figure 7 ijms-23-00410-f007:**
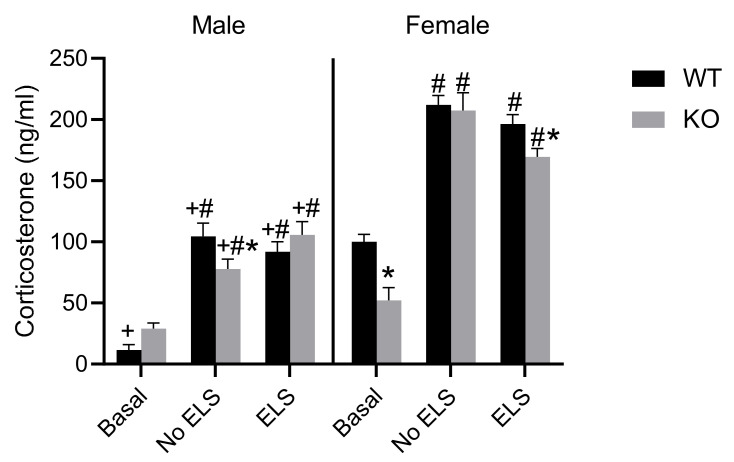
Effect of ELS in WT and GPR37KO mice on corticosterone levels in adulthood. N/group: WT no-ELS male = 5, WT ELS male = 6, KO no-ELS male = 7, KO ELS male = 5, WT no-ELS female = 5, WT ELS female = 7, KO no-ELS female = 7 and KO ELS female = 7. All data represent mean ± SEM. * *p* < 0.05 treatment difference within genotype; ^#^
*p* < 0.05 versus corresponding WT; ^+^
*p* < 0.05, with respect to sex; calculated by three-way ANOVA followed by multiple comparisons using two-stage linear step-up procedure of Benjamini, Krieger and Yekutieli. WT = Wild Type, KO = Knockout, ELS = Early Life Stress.

**Figure 8 ijms-23-00410-f008:**
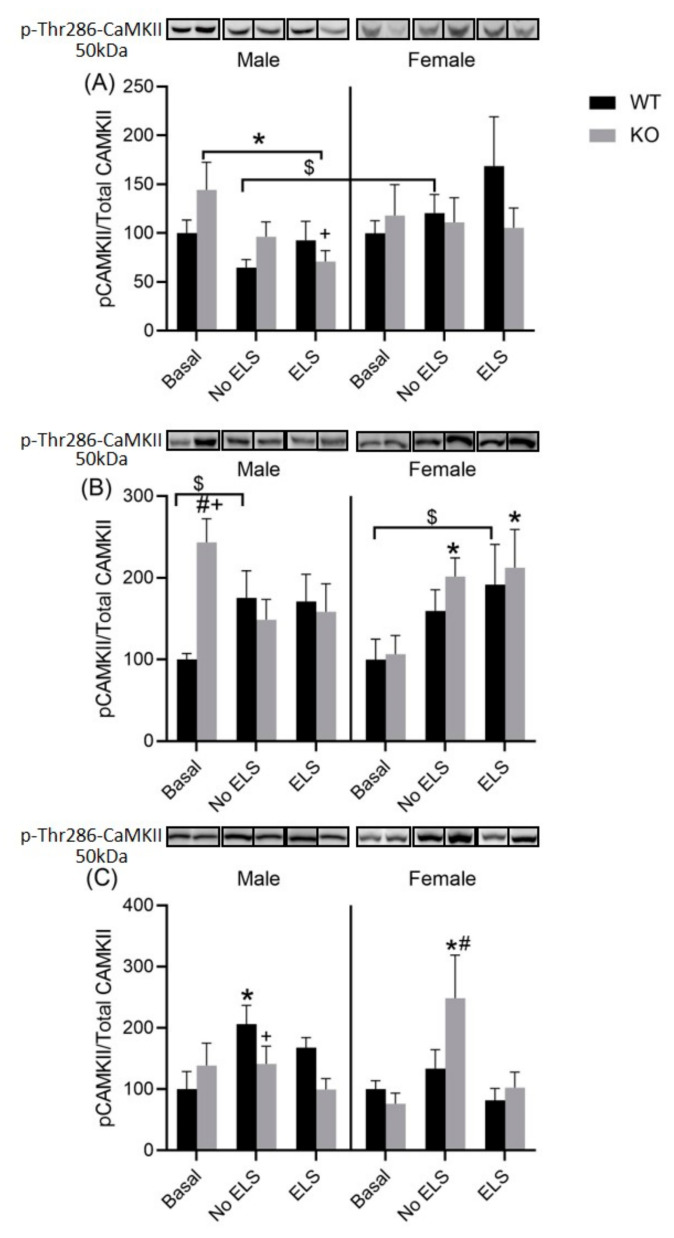
Effect of early life stress in WT and GPR37KO mice on the expression of P-T286-CaMKII/total CaMKII in the dorsal hippocampus (**A**); WT basal male = 6, WT no-ELS male = 7, WT ELS male = 6, KO basal male = 7, KO no-ELS male = 7, KO ELS male = 6, WT basal female = 6, WT no-ELS female = 6, WT ELS female = 6, KO basal female = 6, KO no-ELS female = 6 and KO ELS female = 6. Ventral hippocampus.(**B**); WT basal male = 6, WT no-ELS male = 6, WT ELS male = 7, KO basal male = 7, KO no-ELS male = 7, KO ELS male = 6, WT basal female = 6, WT no-ELS female = 6, WT ELS female = 5, KO basal female = 6, KO no-ELS female = 6 and KO ELS female = 6 and amygdala (**C**) WT basal male = 6, WT no-ELS male = 7, WT ELS male = 6, KO basal male = 7, KO no-ELS male = 6, KO ELS male = 5, WT basal female = 7, WT no-ELS female = 7, WT ELS female = 5, KO basal female = 6, KO no-ELS female = 8 and KO ELS female = 5 All data represent mean ± SEM. * *p* < 0.05 versus corresponding WT; ^#^
*p* < 0.05, treatment difference within genotype; ^+^
*p* < 0.05, with respect to sex; ^$^
*p* > 0.05 < 0.1 calculated by three-way ANOVA followed by multiple comparisons using two-stage linear step-up procedure of Benjamini, Krieger and Yekutieli. WT = Wild Type, KO = Knock Out, ELS = Early Life Stress.

**Table 1 ijms-23-00410-t001:** Effect of early life stress in WT and GPR37KO mice on the expression of total CaMKII in the dorsal hippocampus (DH); WT basal male = 6, WT no-ELS male = 7, WT ELS male = 6, KO basal male = 7, KO no-ELS male = 7, KO ELS male = 6, WT basal female = 6, WT no-ELS female = 6, WT ELS female = 6, KO basal female = 6, KO no-ELS female = 6 and KO ELS female = 6; Ventral hippocampus (VH); WT basal male = 6, WT no-ELS male = 6, WT ELS male = 7, KO basal male = 7, KO no-ELS male = 7, KO ELS male = 6, WT basal female = 6, WT no-ELS female = 6, WT ELS female = 5, KO basal female = 6, KO no-ELS female = 6 and KO ELS female = 6; and Amygdala (Amy); WT basal male = 6, WT no-ELS male = 7, WT ELS male = 6, KO basal male = 7, KO no-ELS male = 6, KO ELS male = 5, WT basal female = 7, WT no-ELS female = 7, WT ELS female = 5, KO basal female = 6, KO no-ELS female = 8 and KO ELS female = 5. All data represent mean ± SEM.

Structure	Gender	Group	Total CaMKII
DH	Male	WT Basal	100 ± 5.3
		KO Basal	115.2 ± 9.6
		WT no ELS	124.8 ± 15.9
		KO no ELS	99.2 ± 5.0
		WT ELS	111.1 ± 12.3
		KO ELS	110.3 ± 8.7
	Female	WT Basal	100 ± 5.7
		KO Basal	87.8 ± 11.4
		WT no ELS	91.0 ± 9.1
		KO no ELS	94.7 ± 5.3
		WT ELS	84.1 ± 14.1
		KO ELS	89.4 ± 7.2
VH	Male	WT Basal	100 ± 11.3
		KO Basal	95.1 ± 16.1
		WT no ELS	90.4 ± 8.8
		KO no ELS	86.8 ± 6.1
		WT ELS	68.0 ± 8.2
		KO ELS	82.2 ± 9.2
	Female	WT Basal	100 ± 15.5
		KO Basal	87.3 ± 4.6
		WT no ELS	81.6 ± 12.1
		KO no ELS	97.8 ± 10.4
		WT ELS	90.3 ± 11.1
		KO ELS	92.0 ± 9.3
Amy	Male	WT Basal	100 ± 13.6
		KO Basal	121.1 ± 26.8
		WT no ELS	90.4 ± 20.7
		KO no ELS	81.5 ± 10.5
		WT ELS	76.2 ± 12.4
		KO ELS	101.6 ± 21.5
	Female	WT Basal	100 ± 9.5
		KO Basal	107.6 ± 20.4
		WT no ELS	144.7 ± 22.9
		KO no ELS	132.6 ± 14.9
		WT ELS	127.8 ± 19.7
		KO ELS	135.7 ± 23.1

**Table 2 ijms-23-00410-t002:** Primers used for GPR37 genotyping.

Primer	Sequence, 5′–3′
GPR37 mutant, forward	GGGTGGGATTAGATAAATGCCTGCTCT
GPR37 WT, forward	AACGGGTCTGCAGATGACTGGGTTC
Common, reverse	GGCCAAGAGAGAATTGGAGATCGTC

## Data Availability

All the data used in this study are available from the corresponding author upon request.

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
