# Peer review of "The Effect of Early Life Stress on Emotional Behaviors in GPR37KO Mice"

_ijms, 2021, doi:10.3390/ijms23010410_

Round 1
Reviewer 1 Report
In their study, Veenit and colleagues analyzed the emotional behavior of mice deficient for GPR37, a substrate of Parkin. The used limited nesting and bedding material as early life stress (ELS) from PND 2 to 9 and tested anxiety-like behavior in an elevated plus maze and a light-dark-box, emotional memory via a passive avoidance task and stress coping with the Porsolt forced swim test in male and female mice ko mice and their wild type litter mates. They furthermore checked the body weight, corticosterone serum levels and relative protein levels of P-Thr286-CaMKII in the dorsal and ventral hippocampus and the amygdala via Western blot. The authors found rather increased anxiety in female WT mice after ELS, while female KO mice after ELS showed anxiety levels comparable to non-stressed WT mice in the EPM and LD test. Altered anxiety-like behavior in male mice was no observed or effects were less consistent throughout the tests. However, during passive avoidance, male wt mice showed a reduced retention latency after ELS that was normalized to wt control levels in KO mice with ELS. Females showed lower training latencies and no differences in the retention latencies. ELS had no effect on the immobility time in the FST in females and induced only a trend in male mice. However, in this test male KO mice had already elevated immobility times compared to their wildtype littermates without any ELS.
Following ELS, body weight was reduced by the stress procedure at P12. The effect was only maintained into adulthood in female WT mice after ELS. ELS had further no long-term impact on corticosterone serum levels, which were elevated 30 min after FST but less pronounced in KO (male especially in the no ELS/ female in the ELS group). Basal corticosterone levels in behaviorally naïve mice were generally higher on females compared to males, but less pronounced in female KO mice compared to WT. Finally, phosphorylation of CaMKII was increased in the ventral hippocampus of male KO mice under basal conditions and in the amygdala of female KO mice after behavioral testing, but without ELS.
While this complex set of results is interesting for evaluating GPR37 as a potential target for affective behavior and affective co-morbidities in Parkinson, there are points regarding data presentation and discussion that need to be clarified and results of the tests performed need to be re-considered more carefully.
- It remains quite elusive why the authors selected the specific phosphorylation side of CaMKII for their study. Other signaling pathways or targets play an important role in plasticity as well (e.g. P-CREB). Is there any connection to Parkin or other know GPR37 interaction partners? Please add the background and rationale for this target to the introduction.
- Results & Fig 1: Please describe the trend indicated in the figure (p<0.1) in the results text or leave out completely. When indicating significances in the graphs, please check carefully the legend. Sex differences should be indicated only above male or female bars but not above both, which is quite confusing. Overall, time in open arm is relatively low. Another important parameter to test here would be the total distance covered and the % distance on open arms, since gross differences in locomotion may corrupt the read-out of this activity sensitive test (and for LD-test, which is based on activity as well). Please check the sentence for grammatical errors: p. 3/ l. 90-92, too.
- Results & Fig. 2+3+5: Please check again the positions of the symbols indicating significant differences. Either the meaning of * and # was switched in this graph or the results in the text do not match the graph. Again, check that sex differences are not indicated double.
- Passive avoidance (Fig. 4) would be again sensitive to locomotion. Re-check therefore the EPM distance, for example. Females seem to have a low training latency. Therefore, check carefully for sex differences in locomotion. Double check symbols for indicating significant differences of sex, ELS, genotype is the graph. The post hoc difference described between male ELS WT vs. KO mice (p. 4/ l. 110-112) is not indicated in the graph (or symbols are switched).
- Whether the forced swim test is really assessing depressive-like behavior is currently under debate (see e.g. Molendijk & deKloet, EJN, 2021, DOI: 1111/ejn.15139), therefore wording should be considered carefully (e.g. test for passive vs. active coping style).
- For corticosterone levels, the text (p. 7, l. 169) indicates higher levels for female basal KO mice, but actually the graph shows lower levels for the KO. It would be also helpful to shortly state what basal and behavioral levels mean for corticosterone already here (e.g. 30 min after last behavioral test, FST). See p. 7/l. 160.
- For P-CaMKII levels in the amygdala of females (Fig. 7C, p. 8/l. 195) the phosphorylation is rather increased in the KO mice, since WT would be the more “natural condition”. Please check and re-phrase.
- For all P-CaMKII levels: Did the authors make sure that the basal CaMKII levels are unaltered between conditions and genes types? Provide the data as a supplement figure or as a table in the manuscript.
- Many points in the discussion and the conclusions drawn from the data should be reconsidered
- Was motor behavior properly controlled/ checked for in GPR37 mice, since many of the behavioral tests depend on exploration? Are there sex differences described for locomotion in GPR37 KO mice?
- 10/l. 214: FST is not the best test for depression. Furthermore, many statements are made while the statistical effect is not seen, e.g. the increased immobility time after ELS in male WT mice is not significant (l. 217) and anxiety-like behavior in males was only changed in 1 out of 4 parameters (and only in the LD, but not the EPM test). Thus, the suggestion that male GPR37 KO mice have higher levels of anxiety (l. 220-21) is not supported by the data and should be reconsidered with care.
- The authors state that the observed ELS effects match previous findings (l. 231), but actually the ELS impact, even on WTs, is rather minor. Please compare with previous findings and discuss why only minor differences are apparent in the current study. Again, while the authors focus a lot on the effects in males (passive avoidance, l. 234), the ELS effect on anxiety and the interaction with the genotype seems much more robust in females across both anxiety tests. A focus on this finding and the possible elevation of the ELS effects in GPR37 seems the most striking finding of the study. The correlation of anxiety levels and high corticosterone would also argue for an increased HPA axis modulation in female WT vs. KO mice. The last statement in the abstract (l. 20) should be reconsidered.
- Recheck also the results of the P-CaMKII levels and their discussion. The authors state differences for the dorsal hippocampus that are not shown in the graphs (l. 259-260) and state that KO females after behavior have lower densities, although they are actually higher (l. 263-264). The described elevating effect of behavioral testing in males in the amygdala (l. 264-266) is not supported by the data, too. Overall, it is not clear why P-CaMKII is important and how it could help to interpret the observed phenotype with the gene and sex differences. Likewise, corticosterone effects have multiple effects and a more detailed discussion on the relation of corticosterone and anxiety as well as sex differences would be favorable.
- Please add some more details to the Materials and Methods section:
- Please add circadian phase of testing or time of lights on/ off in the animal house. Time of testing is indicated from 9:00-16:00 h (p. 12/ l. 324), but the relation to light cycle is not further described.
- The authors state that behavioral testing did not occur in a particular estrous cycle phase, but animals were counterbalanced for their cycle phase (l. 319-321). How was the estrous cycle checked? Did the testing interfere with the behavioral test battery? How was stress induced by estrous testing in females vs. males accounted for?
- Please indicate the exact n for each group. In the figure legends only approx. values for each group are shown.
- The automated analysis of the forced swim test was performed on animals filmed sideways or from above?
- Please indicate how the basal group is treated. Are the animals handled parallel to the behavioral tests or are they left undisturbed in the home cage?
Reviewer 2 Report
In this study the authors investigated GPR37 ko mice in combination with early life stress (ELS). They found several sex specific effects on the emotional behaviour of GPR37 ko with ELS and describe changes in plasma corticosterone levels, and alphaCaMKII phosphorylation in the brain.
The study methodology and data analysis appear sound, and the findings are clearly and concisely described and discussed. Overall, this study and its results are very interesting and clearly merit publication. In only have a some very minor comments as outlined below:
Overall:
- Throughout the authors should be specific about which CaMKII they investigated (i.e. alphaCaMKII)
- When referring to their experiments, instead of saying “animals” the authors should refer to the specific species (i.e. mice)
Introduction:
- The introduction is missing the information on the relevance for PD. A few sentence to this regard should be included as e.g. the abstract and also the discussion refer to this.
- Lines 44, 60 and 62: instead of “gender”, the term “sex” should be used as is done correctly in the rest of the manuscript.
Results:
- Throughout the results section, the authors only indicate significant ANOVA statistics. All ANOVA tests statistics for ALL factors and ALL interactions should be included. If the authors feel that this would disturb the flow of the results section, at least a table or supplementary table containing this information should be included.
- For post hoc results the respective P-values (including the relevant non-significant ones) should be indicated in the results section.
- Line 118 and 134: For the tendency/trend the specific p value should be indicated in the results text. Similarly, instead of writing p<0.1 in the respective figures (i.e. 3B and 4), the specific p-value should be written.
Figures:
- In figure 1A it is not clear what unit is used for “frequency”. This should be clarified (perhaps also using a more appropriate term than frequency; e.g. number of visits)
- The descriptions in the legend for the groups is a bit misleading as “NO” seems like an abbreviation. Perhaps this should be written in small caps.
- Abbreviations used in Figures, e.g. WT, KO, ELS should be described in the figure legends.
- Figure 8: From the current depictions it appears as if behavioural experiments started right after ELS on P10. Perhaps the authors could indicate the time between ELS and onset of experiments in figure and/or somehow graphically emphasize this (e.g. by interrupting the line).
Also the authors should consider placing this figure before all others as Figure 1
Discussion:
- Line 217: There is something wrong with the wording “… animals tended showed better retention…”
- Line 234/235: Could the better retention memory in GPR37KO ELS be interpreted as putting excessive value into emotional stimuli, sth. that is commonly observed in several mental disorders? Perhaps this could be included as an aspect in the discussion?
- Lines 238-249: The authors should clarify somewhere in this section that all mice except the basal one were exposed to the FST 30 min before being sacrificed. Thus, the reduced corticosterone is not surprising. This aspect should then also go into the interpretation, i.e. differences in baseline and acute stress-induced corticosterone levels. The same could be done for P-T-CaMKII
- At the end of the discussion a short section for limitations and a short conclusion should be included.
Materials and Methods:
- Line 313: What is meant by “… all the groups were culled from…” This should be clarified
- Line 321: It should be clarified how estrous cycle was determined, how reliable this is and if this might have caused additional stress.
- Line 330: How was isopentane cooled. With dry ice? This should be clarified.
- Lines 338-340: This statement needs references.
- Lines 338-342: This section is a repetition from section 4.3, i.e. should be shortened/removed in one or the other.
- Section 4.7 and 4.8: Clear identifiers, such as order numbers and/or RRIDs should be included for the used antibodies and ELISA kit.
Author Response
Reviewer 2
Overall:
- Throughout the authors should be specific about which CaMKII they investigated (i.e. alphaCaMKII)
Ans: Thanks to our reviewer for pointing this. We have now corrected it throughout the text.
- When referring to their experiments, instead of saying “animals” the authors should refer to the specific species (i.e. mice)
Ans: We have incorporated the suggestion from our reviewer and have now changed animals to mice.
Introduction:
- The introduction is missing the information on the relevance for PD. A few sentence to this regard should be included as e.g. the abstract and also the discussion refer to this.
Ans: We have now added the link between PD and GPR37KO in context of non-motor behavior in the introduction part line 41-48
- Lines 44, 60 and 62: instead of “gender”, the term “sex” should be used as is done correctly in the rest of the manuscript.
Ans: We have done the correction suggested by our reviewer
Results:
- Throughout the results section, the authors only indicate significant ANOVA statistics. All ANOVA tests statistics for ALL factors and ALL interactions should be included. If the authors feel that this would disturb the flow of the results section, at least a table or supplementary table containing this information should be included.
Ans: We have now added the detail ANOVA analysis and the relevant post hoc information in the supplementary and also elaborated on the statistics in the text.
- For post hoc results the respective P-values (including the relevant non-significant ones) should be indicated in the results section.
Ans: We have now indicated the relevant p values in the text and extended entire relevant p values all the relevant Post hoc p values in the supplementary material and indicated in the text.
- Line 118 and 134: For the tendency/trend the specific p value should be indicated in the results text. Similarly, instead of writing p<0.1 in the respective figures (i.e. 3B and 4), the specific p-value should be written.
Ans: We have now indicated a tendency/trend as $p < 0.1> 0.05
Figures:
- In figure 1A it is not clear what unit is used for “frequency”. This should be clarified (perhaps also using a more appropriate term than frequency; e.g. number of visits)
Ans: We have now changed frequency to number of visits as per reviewer’s suggestion.
- The descriptions in the legend for the groups is a bit misleading as “NO” seems like an abbreviation. Perhaps this should be written in small caps.
Ans: We have now corrected this in the legend as suggested by the reviewer
- Abbreviations used in Figures, e.g. WT, KO, ELS should be described in the figure legends.
Ans: done
- Figure 8: From the current depictions it appears as if behavioural experiments started right after ELS on P10. Perhaps the authors could indicate the time between ELS and onset of experiments in figure and/or somehow graphically emphasize this (e.g. by interrupting the line). Also the authors should consider placing this figure before all others as Figure 1
Ans: We thank you our reviewer for this suggestion. We have now moved figure 8 as figure 1 which provides schematic of the study design
Discussion:
- Line 217: There is something wrong with the wording “… animals tended showed better retention…”
Ans: We have now corrected the wording (due to edits within the text, we are refraining from writing the exact line no)
Line 234/235: Could the better retention memory in GPR37KO ELS be interpreted as putting excessive value into emotional stimuli, sth. that is commonly observed in several mental disorders? Perhaps this could be included as an aspect in the discussion?
Ans: GR37KO animals, particularly males have higher retention in passive avoidance task suggesting they might be better in retaining the memory of aversive stimuli. The point raised by the reviewer is interesting and could be likely.
- Lines 238-249: The authors should clarify somewhere in this section that all mice except the basal one were exposed to the FST 30 min before being sacrificed. Thus, the reduced corticosterone is not surprising. This aspect should then also go into the interpretation, i.e. differences in baseline and acute stress-induced corticosterone levels. The same could be done for P-T-CaMKII
Ans: We have mentioned in materials and methods part (in general experimental plan) explaining the conditions for the basal group and how they were terminated along with animals subjected to behaviour tests. We do agree and the corticosterone difference between basal group vs the groups subjected to behaviour tests are expected.
- At the end of the discussion a short section for limitations and a short conclusion should be included.
Ans: We have included conclusion and limitation section at the end of the manuscript.
Materials and Methods:
- Line 313: What is meant by “… all the groups were culled from…” This should be clarified
Ans: We apologize for the usage of unclear word here. We have now replaced culled with selected in in this edited version
- Line 321: It should be clarified how estrous cycle was determined, how reliable this is and if this might have caused additional stress.
Ans: The estrous cycle was checked with vaginal smears before the first behavior test and the animals were counterbalanced across groups which suggest the readouts in any group could not have been due to the animals being in a different cycle compared to rest of the groups. No, testing did not interfere with the behavioral test battery since, the females were tested just once one day prior to the first behavior test.
- Line 330: How was isopentane cooled. With dry ice? This should be clarified.
Ans: The isopentane was cooled down with dry ice during the experiment. This is added now in the main text.
- Lines 338-340: This statement needs references.
Ans: Thanks to our reviewer for spotting this. We have now added the reference now in the edited version.
- Lines 338-342: This section is a repetition from section 4.3, i.e. should be shortened/removed in one or the other.
Ans: We agree with our reviewer’s comment and now we have shortened section 4.4
- Section 4.7 and 4.8: Clear identifiers, such as order numbers and/or RRIDs should be included for the used antibodies and ELISA kit.
Ans: We have now added clear identifiers in section 4.7 and 4.8
